# Extracellular Matrix-Related Hubs Genes Have Adverse Effects on Gastric Adenocarcinoma Prognosis Based on Bioinformatics Analysis

**DOI:** 10.3390/genes12071104

**Published:** 2021-07-20

**Authors:** Husile Alatan, Yinwei Chen, Jinghua Zhou, Li Wang

**Affiliations:** 1School of Basic Medicine, Hangzhou Normal University, Hangzhou 311121, China; altanhusler@gmail.com (H.A.); becose@stu.hznu.edu.cn (Y.C.); 2NS Bio Japan Co., Ltd., Akita 0130205, Japan; 3Department of Evolutionary Studies of Biosystems, School of Advanced Sciences, Graduate University for Advanced Studies (SOKENDAI), Hayama 2400193, Kanagawa, Japan

**Keywords:** gastric adenocarcinoma, bioinformatics analysis, extracellular matrix, biomarkers

## Abstract

Gastric adenocarcinoma (GAC) is the most frequent type of stomach cancer, characterized by high heterogeneity and phenotypic diversity. Although many novel strategies have been developed for treating GAC, recurrence and metastasis rates are still high. Therefore, it is necessary to screen new potential biomarkers correlated with prognosis and novel molecular targets. Gene expression profiles were obtained from the from NCBI Gene Expression Omnibus (GEO) database. We conduct an integrated analysis using the online Venny website to explore candidate hub genes between differentially expressed genes (DEGs) of two datasets. Gene ontology (GO) and Kyoto Encyclopedia 18 of Genes and Genomes (KEGG) pathway enrichment analysis found that extracellular matrix plays an important role in GAC. In addition, we applied protein-protein interaction (PPI) network analysis by using the Search Tool for the Retrieval of Interacting Genes (STRING) and visualized with Cytoscape software. Furthermore, we employed Cytoscape software to analyze the interactive relationship of candidate gene for further analysis. We found that ECM related proteins played an important role in GAC, and 15 hub genes were extracted from 123 DEGs genes. There were four hub genes (*bgn*, *vcan*, *col1a1* and *timp1*) predicted to be associated with poor prognosis among the 15 hub genes.

## 1. Introduction

Stomach cancer, one of the most common malignancies, is the third leading cause of cancer-related death [1]. As the most frequent type of stomach cancer, gastric adenocarcinoma (GAC) causes a significant public health burden with a 5-year overall survival (OS) rate of less than 30% [2]. There are four subtypes of gastric cancer identified by TCGA Project: tumors positive for Epstein-Barr virus, microsatellite unstable tumors, genomically stable tumours and tumors with chromosomal instability [3]. To date, various genetic factors have been reported to be associated with the pathogenesis of GAC. There are several treatment options for GAC including anti-her2 therapy, anti-VEGF therapy, anti-EGFR therapy and anti-FGFR-2 therapy; however, their efficiency is hampered by toxicities [4]. A number of studies have been conducted to investigate the pathogenesis of GAC. Nevertheless, the exact mechanisms are still not well defined. Using bioinformatics analysis techniques, some potential biomarkers can be found through big database analysis. 

Public bioinformatics databases such as GEO and the cancer genome atlas (TCGA) allow the acquisition of gene expression profiles, which can be followed by functional analysis to screen significant genetic or epigenetic variations occurring in carcinogenesis. Indeed, bioinformatic methods are promising for the screening of potential biomarkers that are feasible for clinical diagnosis and prognosis evaluation. In the past five years, many researchers have studied gastric cancer through multiomics data [5,6,7,8,9]. Through the analysis of multi-omics data, tumor immune signals and microenvironment can be reshaped during neoadjuvant chemotherapy. C10ORF71 mutation affects the efficacy of neoadjuvant chemotherapy for gastric cancer [5]. The efficacy of pembrolizumab in GC patients can be assessed using tumor microenvironment evaluation through multiomics data analysis [6]. The expression of some extracellular matrix (ECM) related genes has been associated with poor prognosis in many cancers [10,11]. Unfortunately, few studies have reported the effects of ECM on the pathogenesis and prognosis of GAC. 

In this study, we investigated the molecular signatures and transcription networks of differentially expressed genes (DEGs). Additionally, we explored the correlation and overall survival (OS) of stomach adenocarcinoma (STAD) based on hub genes of the TCGA database at GEPIA website. The results may provide new light on the biomarkers of GAC derived from bioinformatics analyses.

## 2. Materials and Methods

### 2.1. Microarray Data Collection

Two gene expression profiles (i.e., GSE103236 and GSE96668) were downloaded from the NCBI Gene Expression Omnibus (http://www.ncbi.nlm.nih.gov/geo/, accessed on 27 July 2020). The array data of GSE103236 consisted of 11 cancer samples and nine normal adjacent tissue samples from GAC patients [12]. GSE96668 was consisted of 60 samples including 49 gastro-esophageal adenocarcinoma samples and 11 noncancer control samples [13].

### 2.2. Data Processing

For the data comparison between GSE103236 and GSE96668, an interactive web tool, GEO2R, was utilized based on the R programming language, and we selected the Benjamini & Hochberg (false discovery rate) as the *p*-value adjustment option [14]. The adjusted *p*-value < 0.05 and |log Fold Change (FC)| ≥ 1 were used as the cutoff criteria for statistical analysis of each dataset.

### 2.3. Identification of Coexpression Modules

In order to identify the intersection of nodes among the GSE103236 and GSE96668 DEGs genes, we further employed an online Venn diagram (http://bioinfogp.cnb.csic.es/tools/venny/, accessed on 27 July 2020). As a result, the DEGs, including coupregulated genes and t codownregulated genes, were identified between the two databases.

### 2.4. Functional and Pathway Enrichment Analysis

To better explore the biological significance of these co-DEGs, we performed the Gene ontology (GO) and Kyoto Encyclopedia of Genes and Genomes (KEGG) pathway enrichment analysis by using the online Database for Annotation, Visualization and Integrated Discovery (DAVID; https://david.ncifcrf.gov/summary.jsp, accessed on 27 September 2020). The results of Biological Process (BP), cellular component (CC), molecular function (MF) and KEGG pathway were download and further visualized by using R software 3.6.3 (https://www.r-project.org/, accessed on 27 September 2020).

### 2.5. PPI Network Construction and Analysis

The Search Tool for the Retrieval of Interacting Genes (STRING) (version 11.0) (http://www.string-db.org/, accessed on 27 September 2020) was applied to analyze the network among the DEGs proteins. Then, the PPI networks for the DEGs genes were visualized by Cytoscape software (version 3.8.0). Genes with degrees >10 were selected as hub genes among the PPI networks. The molecular complex detection (MCODE) plugin was used to create the modules in Cytoscape with the following parameters: Degree cutoff of 2, node score cutoff of 0.2, k-core of 3, and max. depth of 100.

### 2.6. The Expression and Survival Analysis of Hub Genes

Gene Expression Profiling Interactive Analysis (GEPIA) (http://gepia.cancer-pku.cn, accessed on 27 September 2020) was utilized to validate the expression of hub genes in TCGA-STAD tumor samples [15]. To further analyze the effects of the hub genes on GAC patient prognosis survival, the GEPIA website was selected to draw overall survival (OS) curves. Meanwhile, we selected quartile as the group cutoff. Genes that significantly affect the prognosis survival of GAC patients were further analyzed for correlation with each other in the GEPIA website. The protein levels of these genes were obtained from the Human Protein Atlas (https://www.proteinatlas.org/, accessed on 26 January 2021).

## 3. Results

### 3.1. DEGs and Clusters

In total, 501 genes were extracted from the GSE103236 dataset and 904 genes were extracted from the GSE96668 datasets, respectively (Figure 1). In the GSE103236 data, 331 genes were upregulated and 170 genes were downregulated. In GSE96668 dataset, 416 genes were upregulated and 488 genes were downregulated. A total of 123 DEGs were identified in the two datasets, including 75 upregulated and 48 downregulated DEGs (Table 1, Figure 2).

### 3.2. Functional and Pathway Enrichment Analyses

The top 10 significant enrichment results of the GO term were classified into three parts: biological process group, molecular function group and cellular component group (Figure 3A–C). A *p*-value < 0.05 was regarded as the threshold value. The enriched GO terms and gene lists with the top 10 significant enrichment results are shown in Table 2. The first three terms sorted by count are marked in black. The biological processes GO categories were mainly involved in cell adhesion, positive regulation of cell proliferation and negative regulation of apoptotic process (Figure 3A). The GO molecular function analysis revealed enrichment in protein binding, calcium ion binding and chromatin binding (Figure 3B). Moreover, in the cellular components category, the co-DEGs were mainly enriched in extracellular exosome, extracellular space and extracellular region (Figure 3C). The top three enriched KEGG pathways of DEGs were the PI3K-Akt signaling pathway, ECM-receptor interaction and focal adhesion (Figure 3D, Table 3).

### 3.3. PPI Network and Module Analyses 

In order to understand the relationship and interaction between the co-DEGs of the two datasets, we used the online STRING website to analyze the PPI network of co-DEGs-encoded proteins, which was visualized by Cytoscape software. Based on the STRING website, 123 DEGs were filtered into the DEGs PPI networks complex consisting of 88 nodes and 230 edges (Figure 4A). Fifteen hub genes with a degree of > 10 were selected from those networks. Subsequently, we performed module analysis by MCODE, a plugin using scoring and finding parameters for the best results, and two clusters (Figure 4B,C) were screened out from the PPI networks of co-DEGs. Eventually, 15 hub genes were included in the two clusters (Table 3).

### 3.4. Hub Genes Survival Analysis

The mRNA expression levels of 15 hub genes were further analyzed on GEPIA website and it was found that those genes were significantly upregulated (Figure 5). The survival situations of 15 hub genes were analyzed by using the GEPIA website. The results indicated that four genes (i.e., *bgn*, *vcan*, *col1a1* and *timp1*) were closely related with poor prognosis (Figure 6) in TCGA-STAD samples. We further analyzed the correlation between the four genes. The outcome illustrated that *bgn* was significantly correlated with *vcan* (R = 0.78), *col1a1* (R = 0.8) and *timp1* (R = 0.68, Figure 7). In gastric cancer tissues where protein expression was detected, the levels of the protein (BGN, VCAN, COL1A1 and TIMP1) were higher than in normal tissues (Figure 8). Therefore, the expression levels and translation levels of these four genes in cancer tissues were higher than those in normal tissues.

## 4. Discussion

The age-standardized 5-year OS of GAC patients is less than 30% due to poor prognosis in the Chinese mainland [16]. Therefore, it is urgent to understand the molecular basis of GAC based on bioinformatics analysis of microarray data. Our results revealed that four ECM related hub genes predicted poor prognosis in GAC based on bioinformatics analysis. Three ECM related hub genes were well correlated with *bgn*. Therefore, *bgn*, *vcan*, *col1a1* and *timp1* associated with the extracellular matrix may serve as new biomarkers for clinical diagnosis and prognosis of GAC.

The immune tumor microenvironment could affect the survival of cancer patients [17]. ECM is a key component of the microenvironment around the cells, playing important roles during embryonic development and organ homeostasis. ECM is common in the abnormal state and reconstituted in diseases such as cancer [18]. In the present study, GO and KEGG pathways enrichment analyses demonstrated that extracellular environment proteins were crucial for GAC cancer maintenance. Among the coexpression module, the top three terms enriched in CC were all related to the extracellular environment. Cell adhesion is an essential process for cell migration. Directional cell migration is initiated by extracellular cues including ECM proteins and mechanical forces [18]. Cell adhesion is regulated by cellular contact inhibition at the earlier stage of the neoplastic process [19]. Moreover, cell adhesion was the term with the largest number related to BP enrichment in our analysis, while calcium ion binding was the term with the largest number and smallest *p*-value in MF enrichment. The ECM-receptor interaction term enriched in the KEGG pathway had the smallest *p*-value. Therefore, ECM proteins may play an important part in GAC, and further studies are required to investigate the specific mechanisms.

Based on the PPI network using the Cytoscape software, we screened 15 hub genes, among which four (i.e., *BGN*, *VCAN*, *COL1A1* and *TIMP1*) were associated with poor prognosis in TCGA-STAD samples. To further analyze the correlation between the four genes, we found that *VCAN*, *COL1A1* and *TIMP1* genes were closely correlated with biglycan (BGN) serving as an important component of ECM protein belonging to the small leucine-rich proteoglycans family [20]. Currently, a number of studies have indicated that the expression level of BGN is significantly higher in tumor tissues than in adjacent normal tissues [21,22,23]. In addition, it could act as an oncogenic by activating the FAK signaling pathway in metastasis of gastric cancer [24]. In the present study, VCAN was well correlated with BGN. As an ECM proteoglycan, VCAN could interact with other ECM components, and be implicated in regulating cell proliferation, differentiation, apoptosis, migration and adhesion in a variety of malignancies. The expression of versican in some tumor cells showed an increase in bladder cancer [25], colon carcinoma [26], ovarian cancer [27] and hepatocellular carcinoma [28]. In a previous study, the density of VCAN expression was stronger in metastatic tumors compared to the primary tumor [29]. Additionally, the expression of versican in tumors was associated with cancer grade and adverse outcome [30]. Collagen type I α 1 (COL1A1) is the pro-α 1 chain of type I collagen protein, as an essential component of the ECM, involved in many biological processes including ECM remodeling, tumor cell adhesion and cell migration [11,31]. Moreover, COL1A1 accelerated intraperitoneal metastasis of ovarian cancer xenograft by intraperitoneally injection in mice [10]. Previous studies indicated that as a part of ECM, TIMP1 could led to accelerated differentiation and hypertrophy of adipocytes, which contributed to the pathogenesis of cancer [32,33]. A recent study showed that high *TIMP1* mRNA was associated with a lower OS in clear cell renal cell carcinoma [34]. This is consistent with our results that a higher TIMP1 level was associated with poor OS in GAC patients. In summary, BGN, VCAN, COL1A1 and TIMP1 may be helpful in revealing pathogenic mechanism of GAC.

## 5. Conclusions

In summary, four hub genes in the two datasets (i.e., GSE103236 and GSE96668) predicted poor prognosis in TCGA-STAD samples. Among these four hub genes, the expression of *VCAN*, *COL1A1* and *TIMP1* was well correlated with BGN. Moreover, all four genes were significantly upregulated in the mRNA level, which plays an important role in the tumor microenvironment. Eventually, our findings suggested that ECM-related proteins including BGN, VCAN, COL1A1 and TIMP1 may serve as potential candidate biomarkers for the detection and prognosis prediction of GAC.

## Figures and Tables

**Figure 1 genes-12-01104-f001:**
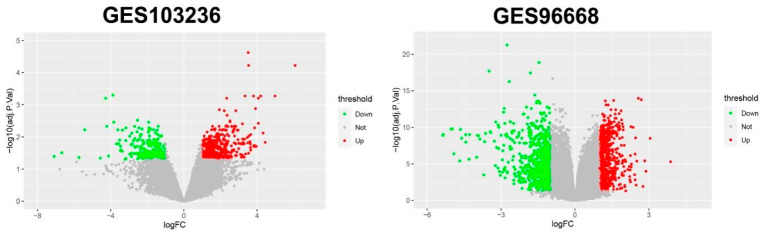
Volcano plot visualizing all the DEGs in GSE103236 and GSE9668. Red dots represent upregulated genes, green dots represent downregulated genes and gray dots represent genes without differential expression.

**Figure 2 genes-12-01104-f002:**
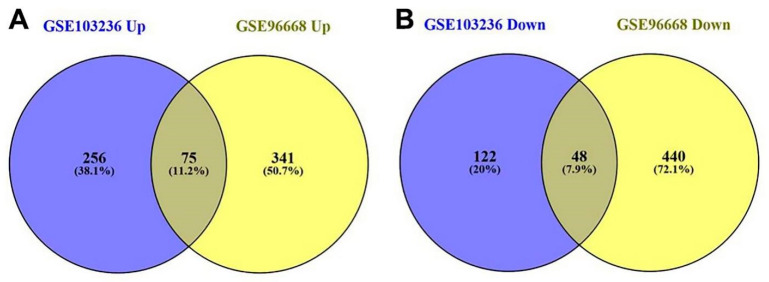
Venn diagram of all screened co-DEGs in GSE103236 and GSE9668. (**A**) Venn diagram of the upregulated co-DEGs. (**B**) Venn diagram of the down-regulated co-DEGs.

**Figure 3 genes-12-01104-f003:**
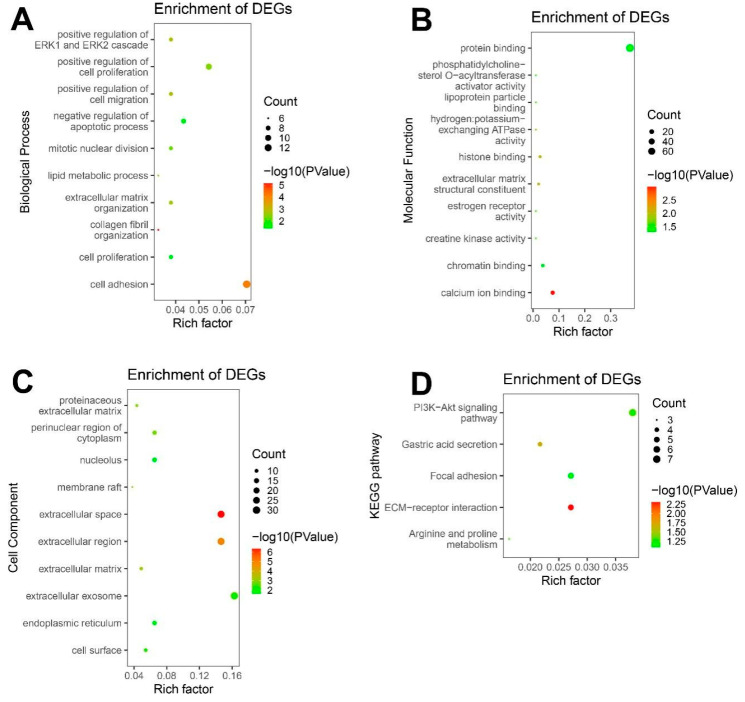
GO functional annotation and KEGG pathway enrichment analysis of co-DEGs. (**A**) The top ten enriched BP of co-DEGs. (**B**) The top ten enriched CC of co-DEGs. (**C**) The top ten enriched MF of co-DEGs. (**D**) The KEGG pathway enrichment of co-DEGs.

**Figure 4 genes-12-01104-f004:**
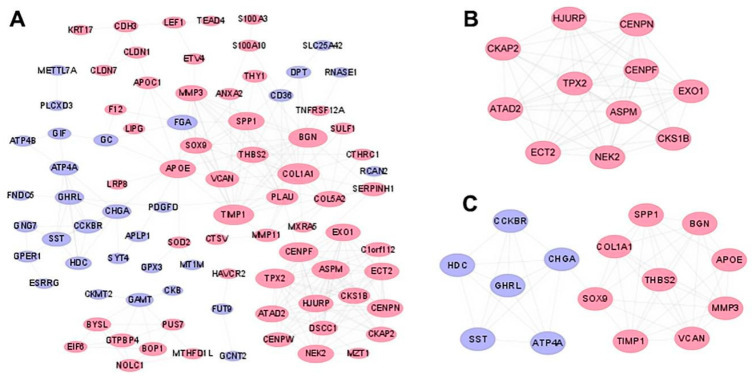
The PPI network and the most important molecular module of co-DEGs. (**A**) The PPI network analyzed by the String tool and visualized by Cytoscape tool. (**B**,**C**) The two modules from the PPI network using the Cytoscape tool. Pink represents the up expression genes. Light blue is the down expression genes.

**Figure 5 genes-12-01104-f005:**
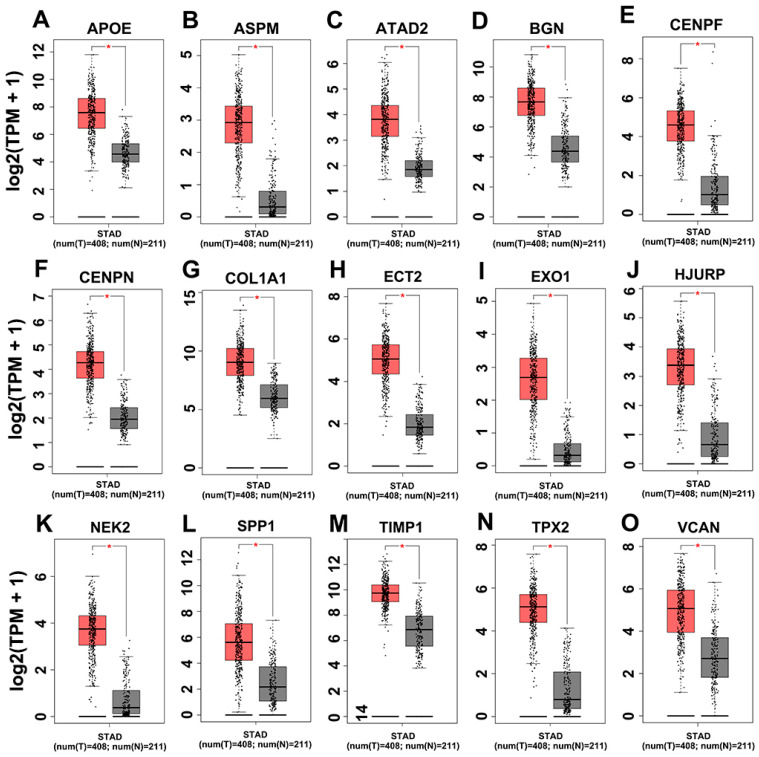
The mRNA expression of 15 hub genes. Validation of the 15 hub genes in the GEPIA box plots showing those genes in mRNA expression using data from the TCGA database and GTEx data in GEPIA. * *p*-values < 0.01. (**A**) APOE; (**B**) ASPM; (**C**) ATAD2; (**D**) BGN; (**E**) CENPF; (**F**) CENPN; (**G**) COL1A1; (**H**) ECT2; (**I**) EXO1; (**J**) HJURP; (**K**) NEK2; (**L**) SPP1; (**M**) TIMP1; (**N**) TPX2; (**O**) VCAN.

**Figure 6 genes-12-01104-f006:**
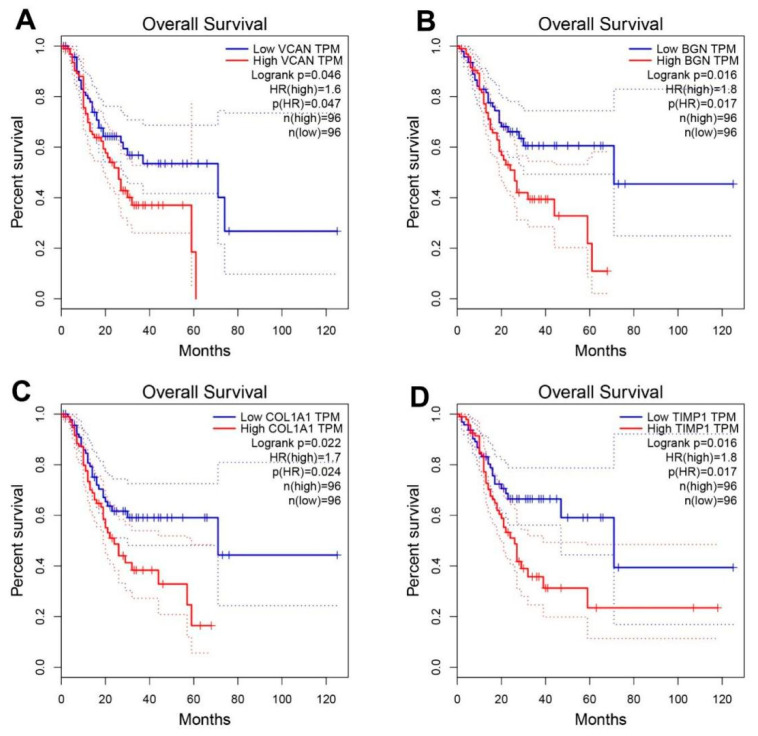
Survival analyses of six hub genes performed using the GEPIA website. Longrank *p* < 0.05 was considered to be statistically significant. (**A**) VCAN; (**B**) BGN; (**C**) COL1A1; (**D**) TIMP1.

**Figure 7 genes-12-01104-f007:**
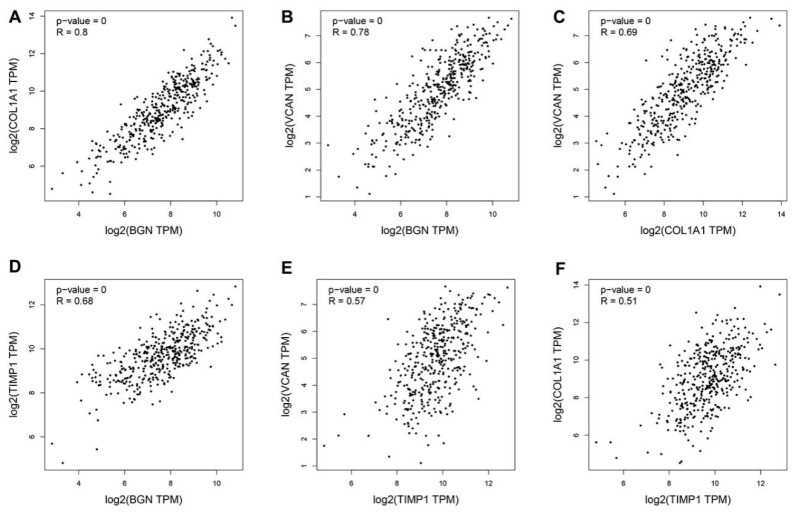
Correlation analyses of four hub genes performed using the GEPIA website. (**A**)The correlation between COL1A1 and BGN. (**B**) The correlation between VCAN and BGN. (**C**) The correlation between VCAN and COL1A1. (**D**) Table 1 and BGN. (**E**)The correlation between VCAN and TIMP1. (**F**) The correlation between COL1A1 and TIMP1.

**Figure 8 genes-12-01104-f008:**
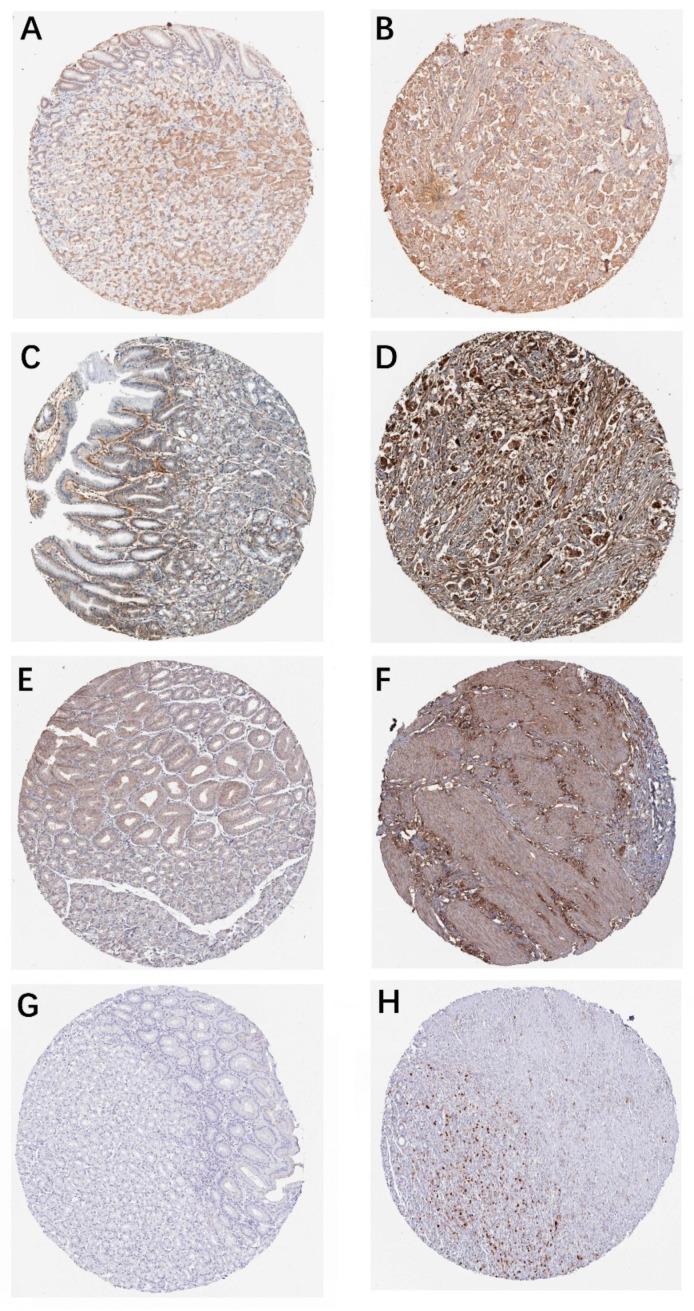
The immunohistochemistry (IHC) of BGN, VCAN, COL1A1 and TIMP1 between normal stomach tissue and cancerous stomach tissue in the human protein atlas. (**A**) The IHC of BGN in normal stomach tissue; (**B**) The IHC of BGN in cancerous stomach tissue; (**C**) The IHC of VCAN in normal stomach tissue; (**D**) The IHC of VCAN in cancerous stomach tissue; (**E**) The IHC of COL1A1 in normal stomach tissue; (**F**) The IHC of COL1A1 in cancerous stomach tissue; (**G**) The IHC of TIMP1 in normal stomach tissue; (**H**) The IHC of TIMP1 in cancerous stomach tissue.

**Table 1 genes-12-01104-t001:** Screening of co-differentially expressed genes of GSE103236 and GSE96668 datasets.

DEGs	List of Gene Symbols
Upregulated DEGs	ANXA2, APOC1, APOE, ASPM, ATAD2, AUNIP, BGN, BOP1, BYSL, C1orf112, CDH3, CEMIP, CENPF, CENPN, CENPW, CKAP2, CKS1B, CLDN1, CLDN7, COL1A1, COL5A2, CTHRC1, CTSV, DSCC1, ECT2, EIF6, ESM1, ETV4, EXO1, F12, FAM72D, FOXD2, GJB2, GTPBP4, HAVCR2, HJURP, HMGA1, IGF2BP3, KRT17, LEF1, LIPG, LRFN4, LRP8, MMP11, MMP3, MTHFD1L, MXRA5, MZT1, NEK2, NFE2L3, NOLC1, NPM3, PLAU, PMEPA1, PUS7, RIPK2, S100A10, S100A3, SEH1L, SERPINH1, SNX10, SOD2, SOX9, SPP1, SULF1, TEAD4, THBS2, THY1, TIMP1, TMEM158, TNFRSF12A, TPX2, UPP1, VCAN, ZFAS1
Downregulated DEGs	ADHFE1, APLP1, APOBEC2, ARHGEF37, ATP4A, ATP4B, C16orf89, C2orf40, CCKBR, CD36, CHGA, CKB, CKMT2, DPT, ESRRG, FGA, FNDC5, FUT9, GAMT, GC, GCNT2, GHRL, GIF, GNG7, GPER1, GPX3, HDC, LIFR, MAL, METTL7A, MT1M, MYRIP, PDGFD, PLCXD3, PNPLA7, PPP2R3A, RCAN2, RERGL, RGN, RNASE1, RPRM, SIGLEC11, SLC25A4, SLC25A42, SLC2A12, SORBS2, SST, SYT4

**Table 2 genes-12-01104-t002:** GO terms functional enrichment analysis of codifferentially expressed genes.

Category	Term	Count	*p* Value
Biological Processes	GO:0007155~cell adhesion	13	7.92 × 10^−5^
	GO:0008284~positive regulation of cell proliferation	10	5.02 × 10^−3^
	GO:0043066~negative regulation of apoptotic process	8	3.84 × 10^−2^
	GO:0070374~positive regulation of ERK1 and ERK2 cascade	7	1.35 × 10^−3^
	GO:0030335~positive regulation of cell migration	7	1.75 × 10^−3^
	GO:0030198~extracellular matrix organization	7	2.40 × 10^−3^
	GO:0007067~mitotic nuclear division	7	7.52 × 10^−3^
	GO:0008283~cell proliferation	7	4.15 × 10^−2^
	GO:0030199~collagen fibril organization	6	6.88 × 10^−6^
	GO:0006629~lipid metabolic process	6	4.68 × 10^−3^
Molecular Function	GO:0005515~protein binding	69	4.98 × 10^−2^
	GO:0005509~calcium ion binding	14	1.01 × 10^−3^
	GO:0003682~chromatin binding	7	4.77 × 10^−2^
	GO:0042393~histone binding	5	9.13 × 10^−3^
	GO:0005201~extracellular matrix structural constituent	4	1.03 × 10^−2^
	GO:0008900~hydrogen:potassium-exchanging ATPase activity	2	1.99 × 10^−2^
	GO:0030284~estrogen receptor activity	2	3.30 × 10^−2^
	GO:0071813~lipoprotein particle binding	2	3.30 × 10^−2^
	GO:0004111~creatine kinase activity	2	3.95 × 10^−2^
	GO:0060228~phosphatidylcholine-sterol O-acyltransferase activator activity	2	3.95 × 10^−2^
Cellular component	GO:0070062~extracellular exosome	30	7.08 × 10^−3^
	GO:0005615~extracellular space	27	4.46 × 10^−7^
	GO:0005576~extracellular region	27	1.20 × 10^−5^
	GO:0048471~perinuclear region of cytoplasm	12	2.49 × 10^−3^
	GO:0005783~endoplasmic reticulum	12	1.96 × 10^−2^
	GO:0005730~nucleolus	12	2.45 × 10^−2^
	GO:0009986~cell surface	10	9.16 × 10^−3^
	GO:0031012~extracellular matrix	9	7.28 × 10^−4^
	GO:0005578~proteinaceous extracellular matrix	8	1.90 × 10^−3^
	GO:0045121~membrane raft	7	2.34 × 10^−3^
	GO:0005581~collagen trimer	6	3.41 × 10^−4^

**Table 3 genes-12-01104-t003:** KEGG pathway enrichment analysis of codifferentially expressed genes.

Category	Term	Count	*p* Value
KEGG PATHWAY	hsa04971:Gastric acid secretion	4	2.03 × 10^−2^
	hsa04512:ECM-receptor interaction	5	4.98× 10^−3^
	hsa04151:PI3K-Akt signaling pathway	7	5.59 × 10^−2^
	hsa00330:Arginine and proline metabolism	3	6.02 × 10^−2^
	hsa04510:Focal adhesion	5	8.17 × 10^−2^

## Data Availability

Data is available at NCBI GEO, accession numbers: GSE103236, GSE96668.

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
