# Peer review of "Extracellular Matrix-Related Hubs Genes Have Adverse Effects on Gastric Adenocarcinoma Prognosis Based on Bioinformatics Analysis"

_genes, 2021, doi:10.3390/genes12071104_

Round 1

Reviewer 1 Report

The manuscript concerns a significant health problem, gastric cancer and the unsatisfactory situation that target therapies are not leading to better outcomes. Work centres on a comparison of global expression analysis between different gastric cancers and either their normal counterparts or controls to identify potential new targets for therapy. The assumption is that differences in expression represent functional importance. Although this manuscript describes analysis leading to the conclusion that differences in the expression of ECM and HUB genes are ‘drivers’ of aggressive disease and therefore potential targets for new therapies, there are a number of limitations of the manuscript and the experiments that need to be addressed.

The first point is that a fluent English Language speaker needs to advise on presentation. A relatively large number of words and phrases that are used are either not good English or lead the reader to be informed inappropriately. For example, the abstract line 18 says “two differentially expressed genes”, but actual means two sets of data. Also, line 33, “great threats to public health” suggests an active challenge whereas it refers to “a significant public health burden”, which is passive and a matter of fact. Line 94 suggests that candidates affect outcome, whereas the analysis merely tests for association, with no suggestion that functional consequences are implied. Many more instances, too many to list are present.

Concerning the science, the conclusions are in part unsubstantiated, either by design or by omission. The following is not a comprehensive list of issues but represents specific issues and topics that lead to my overall disquiet.

The merits of performing a global analysis are advocated but, the focus is on ECM and hub genes. This is essentially a candidate gene approach and moves away from the benefits of non a prior screening, as would be the outcome of a truly global approach.

Two expression data sets are combined for analysis, but they have different controls, which inevitably will increase variance and reduce power. It is therefore not clear whether there is sufficient power in the study to detect significant differences. Cut offs of 0.05 or 0.01 are used but multiple testing with large numbers of genes will inevitably lead to false leads and they may not be significant. Focusing on a specific set of genes, as occurred here, then becomes a self-fulfilling prophesy. A Power Calculation should therefore be performed to determine at what level probability values should be considered significant. In addition, a Bonferroni Correction should be applied to determine how many candidates are likely to be true candidates.

There is also an issue with the lack of detail about the cases used. Gastric cancer is considered to have at least 4 molecular subtypes, each with different prognosis. Multiple grades and stages are also possible. Without knowing the nature of the cases used, it is not possible to know to what extent the results may have been adversely affected by case selection. Similarly, the tumour cellularity of the cases is also not included. Instances of high and low tumour cellularity could be present, and these may be biasing the results. Only 4 instances of in situ data are shown in Figure 8 and no summary across cases or significance is shown.

The 4 molecular subtypes described in the literature were not mentioned in the manuscript reflecting that the literature review is incomplete or inappropriate. There are at least 5 multi-omics studies of gastric cancer in the literature, several in very high impact journals, all of which are relevant to this manuscript, and none have been included. A more thorough and pertinent literature review is therefore needed.

Significant emphasis is placed on functional significance of the analysis, including network analysis. However, expression analysis is based on mRNA levels and the fold changes are not reported. Fold changes could be very small, which may not be significant and may not lead to differences in protein expression and therefore may be of no consequence. Figure 5, which refers to selected instances of supposed value lacks a y axis label, so it is not possible to know what the numbers mean and therefore, whether they could be important. Alternative splicing, which may also be functionally important could also have occurred and this was not include in the comparisons.

Focusing on expression is a missed opportunity concerning function. Larger, multi-omics data sets, including more samples, could have been used. These include mutational data, for which it is much easier and more reliable to assign functional consequences because of the known effects on proteins and known precedents. It would have been far more convincing if mutations with functional consequences were significantly enriched in the pathways selected for study.

Author Response

Response to Reviewer 1 Comments

The manuscript concerns a significant health problem, gastric cancer and the unsatisfactory situation that target therapies are not leading to better outcomes. Work centres on a comparison of global expression analysis between different gastric cancers and either their normal counterparts or controls to identify potential new targets for therapy. The assumption is that differences in expression represent functional importance. Although this manuscript describes analysis leading to the conclusion that differences in the expression of ECM and HUB genes are ‘drivers’ of aggressive disease and therefore potential targets for new therapies, there are a number of limitations of the manuscript and the experiments that need to be addressed.

Point 1: The first point is that a fluent English Language speaker needs to advise on presentation. A relatively large number of words and phrases that are used are either not good English or lead the reader to be informed inappropriately. For example, the abstract line 18 says “two differentially expressed genes”, but actual means two sets of data. Also, line 33, “great threats to public health” suggests an active challenge whereas it refers to “a significant public health burden”, which is passive and a matter of fact. Line 94 suggests that candidates affect outcome, whereas the analysis merely tests for association, with no suggestion that functional consequences are implied. Many more instances, too many to list are present.

Response 1: Thank you for pointing this out. we asked an expert in English to carefully read our manuscript, and based on the feedback from her and our rethinking and rereading of multiple revised drafts, we made The changes we made are marked in blue in our manuscript. The revised parts are as follows: We replaced the "two differentially expressed genes" with "differentially expressed genes (DEGs) of two datasets"(line 14).We replaced the "great threats to public health" with "a significant public health burden"(line 29). The cutoff of in the website of GEPIA is a commonly used data analysis method and many published papers have used this website method [1-3]. Therefore, we used he website of GEPIA methods to analysis the data of our manuscript.

Point 2: Concerning the science, the conclusions are in part unsubstantiated, either by design or by omission. The following is not a comprehensive list of issues but represents specific issues and topics that lead to my overall disquiet.

Response 2: Indeed, the conclusions of our manuscript are in part unsubstantiated.  Further confirmation of the manuscript conclusions is a long-term and complex process, which requires cell experiments, animal experiments and clinical experiments.  And we plan to further study the specific mechanism later.

Point 3: The merits of performing a global analysis are advocated but, the focus is on ECM and hub genes. This is essentially a candidate gene approach and moves away from the benefits of non a prior screening, as would be the outcome of a truly global approach.

Response 3: Our purpose is to screen for potential functional biomarkers of gastric adenocarcinoma. In our GEO clustering analysis, many genes were found enriched in ECM. Through further analysis, the functional of the hub genes were indeed related to ECM. Our subsequent analysis was used the co-differentially expressed genes not used the ECM related genes.

Point 4: Two expression data sets are combined for analysis, but they have different controls, which inevitably will increase variance and reduce power. It is therefore not clear whether there is sufficient power in the study to detect significant differences. Cut offs of 0.05 or 0.01 are used but multiple testing with large numbers of genes will inevitably lead to false leads and they may not be significant. Focusing on a specific set of genes, as occurred here, then becomes a self-fulfilling prophesy. A Power Calculation should therefore be performed to determine at what level probability values should be considered significant. In addition, a Bonferroni Correction should be applied to determine how many candidates are likely to be true candidates.

Response 4: Each data was analyzed separately to screen for differential expressed genes, and then the two groups of differential expressed genes were combined for further analysis. This reduces errors due to environmental conditions and instrument factors. We used adjusted P-value <0.05 and |log Fold Change (FC)| ≥1 as the cutoff criteria for statistical analysis of each dataset. The Bonferroni Correction method of analysis was not chosen in our manuscript. And we selected the “Benjamini & Hochberg” false discovery rate (FDR) method to analysis the datasets of our manuscript [4-5]. The analytical method we adopted is one of the commonly used analytical methods.

Point 5: There is also an issue with the lack of detail about the cases used. Gastric cancer is considered to have at least 4 molecular subtypes, each with different prognosis. Multiple grades and stages are also possible. Without knowing the nature of the cases used, it is not possible to know to what extent the results may have been adversely affected by case selection. Similarly, the tumour cellularity of the cases is also not included. Instances of high and low tumour cellularity could be present, and these may be biasing the results. Only 4 instances of in situ data are shown in Figure 8 and no summary across cases or significance is shown.

Response 5: It is a very good research idea to further study the gastric cancer between different subtypes. Our future research will focus on the different subtypes of gastric cancer. While the examples of high and low tumor cells may skew the results, the large sample size data can correct for this weakness. So we combined these two big datasets for analysis to reduce the bias. After a series of bioinformatics analyses, only 4 genes were screened out, so only 4 in situ data instances were shown in Figure 8. And the Figure 8 is to show that the gene in the cancer group compared to the control group, the gene in the protein translation level is higher. So there was a few summaries across cases in line 163-166.

Point 6: The 4 molecular subtypes described in the literature were not mentioned in the manuscript reflecting that the literature review is incomplete or inappropriate. There are at least 5 multi-omics studies of gastric cancer in the literature, several in very high impact journals, all of which are relevant to this manuscript, and none have been included. A more thorough and pertinent literature review is therefore needed.

Response 6: As the reviewer suggested, we add 4 molecular subtypes described in our manuscript “There are four subtypes of gastric cancer identified by TCGA Project: tumours positive for Epstein-Barr virus, microsatellite unstable tumours, genomically stable tumours and tumours with chromosomal instability [3]” (line 30-33). And we add multi-omics studies of gastric cancer in this manuscript “ In the past five years, many researchers have studied gastric cancer through multi-omics data [5-9]. Through the analysis of multi-omics data, tumor immune signals and microenvironment can be reshaped during neoadjuvant chemotherapy, and C10ORF71 mutation affects the efficacy of neoadjuvant chemotherapy for gastric cancer [5]. The efficacy of pembrolizumab in GC patients can be assessed using tumor microenvironment evaluation through multi-omics data analysis [6]. ” (line 45-52).

Point 7: Significant emphasis is placed on functional significance of the analysis, including network analysis. However, expression analysis is based on mRNA levels and the fold changes are not reported. Fold changes could be very small, which may not be significant and may not lead to differences in protein expression and therefore may be of no consequence. Figure 5, which refers to selected instances of supposed value lacks a y axis label, so it is not possible to know what the numbers mean and therefore, whether they could be important. Alternative splicing, which may also be functionally important could also have occurred and this was not include in the comparisons.

Response 7: The adjusted P-value <0.05 and |log Fold Change (FC)|≥1 were used as the cutoff criteria for statistical analysis of each dataset. Therefore, the data used for subsequent analysis were all greater than or equal to 1 in our manuscript. As reviewer suggested, we added the y axis label in Figure 5 and the y axis is log2(TPM+1). Selective splicing plays an important role in cancer, and we will focus on that in future studies.

Point 8: Focusing on expression is a missed opportunity concerning function. Larger, multi-omics data sets, including more samples, could have been used. These include mutational data, for which it is much easier and more reliable to assign functional consequences because of the known effects on proteins and known precedents. It would have been far more convincing if mutations with functional consequences were significantly enriched in the pathways selected for study.

Response 8: Multi-omics joint study and large sample size can indeed improve the accuracy of research results. Therefore, we will further add multi-omics research in subsequent studies, and the sample size will also be further increased. In the following studies, we will also add functional verification in terms of mutations and other mechanisms.

Reference:

  1. CW Li, ZF Tang, WJ Zhang,et.al, GEPIA2021: integrating multiple deconvolution-based analysis into GEPIA, Nucleic Acids Research, 2021, 49( W1): W242–W246
  2. XW Ma, XQ Wang, Q Dong et.al, Inhibition of KIF20A by transcription factor IRF6 affects the progression of renal clear cell carcinoma, CANCER CELL INTERNATIONAL, 2021, 21(1): 246
  3. Tummanatsakun D, Proungvitaya T , Roytrakul S, et. al, Bioinformatic Prediction of Signaling Pathways for Apurinic/Apyrimidinic Endodeoxyribonuclease 1 (APEX1) and Its Role in Cholangiocarcinoma Cells, 2021, 26(9): 2587
  4. Benjamini, Y., and Hochberg, Y.Controlling the false discovery rate: a practical and powerful approach to multiple testing. Journal of the Royal Statistical Society Series B, 1995, 57, 289-300.
  5. Benjamini, Y., and Yekutieli, D. (2001). The control of the false discovery rate in multiple testing under dependency. Annals of Statistics, 2001, 29, 1165-1188.

Reviewer 2 Report

The present manuscript titled "Extracellular matrix-related hubs genes have adverse effects on gastric adenocarcinoma prognosis based on bioinformatics analysis" tries to predict genes associated with poor prognosis of gastric adenocarcinoma using bioinformatic analysis. The authors need to address the following comments/questions.

Major comments:

  1. What was the FDR threshold? The authors should calculate the FDR for their hits and select hits with no higher than a 5% FDR threshold.
  2. Was the IHC staining in figure 8 performed by the authors? There's no description or reference in the methods or figure legend if the figure was derived from another study?
  3. The manuscript requires grammatical corrections, some of which are mentioned below.

Minor comments:

  1. Abstract: replace "frequency type" with "frequent type"
  2. Abstract: replace "Gene ontology (GO) and Kyoto Encyclopedia 18 of Genes and Genomes (KEGG) pathway enrichment analysis and found that" with "Gene ontology (GO) and Kyoto Encyclopedia 18 of Genes and Genomes (KEGG) pathway enrichment analysis found that"
  3. Even though common, please use full forms for TCGA and STAD before inserting their acronyms in the text.
  4. Line 137: replace "In order tounderstand" with "In order to understand"
  5. Line 185: replace "the top three term enriched" with "the top three terms enriched"
  6. Line 216: replace "Moreover, COL1A1 was accelerated" with "Moreover, COL1A1 accelerated"

Author Response

Response to Reviewer 2 Comments

The present manuscript titled "Extracellular matrix-related hubs genes have adverse effects on gastric adenocarcinoma prognosis based on bioinformatics analysis" tries to predict genes associated with poor prognosis of gastric adenocarcinoma using bioinformatic analysis. The authors need to address the following comments/questions.

Major comments:

Point 1: What was the FDR threshold? The authors should calculate the FDR for their hits and select hits with no higher than a 5% FDR threshold.

Response 1: I'm really sorry that I did not write down some specify analysis method in the manuscript. didn't write down the analysis method clearly in the manuscript. The GO2R analytical method provides several P-value adjustment options. We selected the Benjamini & Hochberg”false discovery rate (FDR) method to analysis the datasets of our manuscript. Because FDR is the most commonly used adjustment for microarray data and provides a good balance between discovery of statistically significant genes and limitation of false positives. And we added the analysis method “and we selected the Benjamini & Hochberg (false discovery rate) as the P-value adjustment options [14]” in line 68-69.

Point 2: Was the IHC staining in figure 8 performed by the authors? There's no description or reference in the methods or figure legend if the figure was derived from another study?

Response 2: The IHC staining in figure 8 was obtained from the human protein atlas. We have a bit of description in our method in section 2.3 “And the protein levels of those genes were obtained from the Human Protein Atlas (https://www.proteinatlas.org/) ” (line100-101 ). In order to more clearer, we added the description in figure 8 legend “in the human protein atlas. A: The IHC of BGN in stomach normal tissue; B: The IHC of BGN in stomach cancer tissue; C: The IHC of VCAN in stomach normal tissue; D: The IHC of VCAN in stomach cancer tissue; E: The IHC of COL1A1 in stomach normal tissue; F: The IHC of COL1A1 in stomach cancer tissue; G: The IHC of TIMP1 in stomach normal tissue; H: The IHC of TIMP1 in stomach cancer tissue. ”( line 176-179).

The manuscript requires grammatical corrections, some of which are mentioned below.

Minor comments:

Point 3: Abstract: replace "frequency type" with "frequent type"

Response 3: As reviewer suggested, we replaced the "frequency type" with "frequent type".

Point 4: Abstract: replace "Gene ontology (GO) and Kyoto Encyclopedia 18 of Genes and Genomes (KEGG) pathway enrichment analysis and found that" with "Gene ontology (GO) and Kyoto Encyclopedia 18 of Genes and Genomes (KEGG) pathway enrichment analysis found that"

Response 4: As reviewer suggested, we replaced the "Gene ontology (GO) and Kyoto Encyclopedia 18 of Genes and Genomes (KEGG) pathway enrichment analysis and found that" with "Gene ontology (GO) and Kyoto Encyclopedia 18 of Genes and Genomes (KEGG) pathway enrichment analysis found that".

Point 5: Even though common, please use full forms for TCGA and STAD before inserting their acronyms in the text.

Response 5: As reviewer suggested, we added the full forms of TCGA (line41 ) and STAD (line 56 ). 

Point 6: Line 137: replace "In order tounderstand" with "In order to understand"

Response 6: As reviewer suggested, we replaced the "In order tounderstand" with "In order to understand".

Point 7: Line 185: replace "the top three term enriched" with "the top three terms enriched"

Response 7: As reviewer suggested, we replaced the "the top three term enriched" with "the top three terms enriched".

Point 8: Line 216: replace "Moreover, COL1A1 was accelerated" with "Moreover, COL1A1 accelerated"

Response 8: As reviewer suggested, we replaced the "Moreover, COL1A1 was accelerated" with "Moreover, COL1A1 accelerated".

Round 2

Reviewer 1 Report

The authors have addressed a significant number of my main concerns, primarily the existing literature by adding appropriate references in context and the potential for false discovery. Use of English has improved. It could be improved further but it doesn't actually affect the sense of the paper. Sample size and the extent of in situ data are essentially unchanged.

Reviewer 2 Report

The manuscript is acceptable for publication.